# Rectus Femoris Cross-Sectional Area and Phase Angle asPredictors of 12-Month Mortality in Idiopathic Pulmonary Fibrosis Patients

**DOI:** 10.3390/nu15204473

**Published:** 2023-10-22

**Authors:** Rocío Fernández-Jiménez, Eva Cabrera Cesar, Ana Sánchez García, Francisco Espíldora Hernández, Isabel M. Vegas-Aguilar, Maria del Mar Amaya-Campos, Isabel Cornejo-Pareja, Patricia Guirado-Peláez, Victor Simón-Frapolli, Mora Murri, Lourdes Garrido-Sánchez, Alvaro Martínez Mesa, Lorena Piñel-Jimenez, Miguel Benítez-Cano Gamonoso, Lara Dalla-Rovere, Maria García Olivares, Jose Luis Velasco-Garrido, Francisco Tinahones-Madueño, José Manuel García-Almeida

**Affiliations:** 1Department of Endocrinology and Nutrition, Virgen de la Victoria University Hospital, 29010 Malaga, Spain; rociofernandeznutricion@gmail.com (R.F.-J.); isabel.mva13@gmail.com (I.M.V.-A.); mariadelmarac2@gmail.com (M.d.M.A.-C.); pguirado1991@gmail.com (P.G.-P.); victorsimonfrapolli.med@gmail.com (V.S.-F.); jgarciaalmeida@gmail.com (J.M.G.-A.); 2IBIMA, Málaga Biomedical Research Institute and BIONAND Platform, 29010 Malaga, Spain; anaasanchez12@gmail.com (A.S.G.); moramurri@gmail.com (M.M.); lourgarrido@gmail.com (L.G.-S.); mery.garcia.96@gmail.com (M.G.O.); 3Department of Medicine and Dermatology, Málaga University, 29016 Malaga, Spain; lara92net@gmail.com; 4Department of Endocrinology and Nutrition, Quironsalud Málaga Hospital, Av. Imperio Argentina, 29004 Malaga, Spain; 5Department of Neumology, Virgen de la Victoria University Hospital, 29010 Malaga, Spain; alvaro.marti.91@gmail.com (A.M.M.); lorenapinel@gmail.com (L.P.-J.); miguel.benitezcano.sspa@juntadeandalucia.es (M.B.-C.G.); jlvelascogarrido@hotmail.com (J.L.V.-G.); 6Department of Neumology, Regional University Hospital, 29010 Malaga, Spain; fespildorahernandez@gmail.com; 7Department of Endocrinology and Nutrition, Hospital Universitario Virgen de la Victoria, CIBEROBN, Carlos III Health Institute (ISCIII), University of Málaga, 29016 Malaga, Spain; fjtinahones@hotmail.com; 8Heart Area Clinical Management Unit, Virgen de la Victoria University Hospital, 29010 Malaga, Spain; 9Department of Endocrinology and Nutrition, Carlos de Haya Regional University Hospital, 29010 Malaga, Spain

**Keywords:** idiopathic pulmonary fibrosis, phase angle, nutritional ultrasound, morphofunctional assessment, diseased related malnutrition, mortality

## Abstract

Background: The value of the phase angle (PhA), measured via bioelectrical impedance analysis (BIA), could be considered a good marker of the cell mass and the cell damage of a patient; however, there are new techniques, such as muscle ultrasonography, that allow the quantity and quality of muscle to be assessed in a minimally invasive way. The aim of this study is to determine the prognostic value of morphofunctional techniques in the prognosis of mortality in patients with idiopathic pulmonary fibrosis (IPF). Methods: This multicenter, cross-sectional study included 86 patients with idiopathic pulmonary fibrosis with a mean age of 71 years, 82.7% of whom were male. The nutritional risk of the patients was assessed by means of questionnaires, such as the Subjective Global Assessment (SGA), and non-invasive functional techniques, including BIA, nutritional ultrasound, and hand grip strength (HGS). Statistical analysis of the sample was performed using JAMOVI version 2.3.22. Results: Correlations were made between the RF-CSA techniques with PhA (r = 0.48, *p* < 0.001), BCM (r = 0.70, *p* < 0.001), SMI (r = 0.64, *p* < 0.001), and HGS (r = 0.54, *p* < 0.001). The cut-off points for 12-month mortality were PhA = 4.5° (AUC = 0.722, sensitivity of 72.7% and specificity of 66.6%), BCM = 28.8 kg (AUC = 0.609, sensitivity of 32.4% and specificity of 100.0%), RF-CSA = 3.00 cm^2^ (AUC = 0.857, sensitivity of 64.4% and specificity of 100.0%), 6MMW = 420 m (AUC = 0.830, sensitivity of 63.27% and specificity of 100.0%), and TUG = 7.2 s (AUC = 0.771, sensitivity of 100.0% and specificity of 56.67%). In addition, a multivariate analysis was performed with RF-CSA, HR = 8.11 (1.39–47.16, *p* = 0.020), and PhA of 6.35 (1.29–31.15, *p* = 0.023), taking into account age, sex, and BMI to determine mortality. Finally, a Kaplan–Meier survival analysis was conducted with low or normal values for classical parameters (GAP and T6MM) and new parameters (PhA, BCM, RF-CSA, and TUG). Conclusion: RF-CSA and PhA were shown to be good prognostic markers of 12-month mortality and could, therefore, be useful screening tools to complement the nutritional assessment of IPF patients.

## 1. Introduction

Idiopathic pulmonary fibrosis (IPF) is an interstitial fibrosing disease with an unknown etiology, rapid progression, and a poor prognosis [1].

The management and follow-up of patients with this disease are complex, as they present different comorbidities. Some, such as pulmonary hypertension, emphysema, lung cancer, or gastroesophageal reflux, have been widely studied [2]. Less is known about endocrine disorders. However, a low body mass index (BMI) and weight loss have been associated with increased mortality [3,4]. After the year 2000, the age-standardized death rate for IPF ranged from 0.5 to 12 per 100,000 population per year [5]. Future therapeutic advances after 2010 may contribute to increasing survival trends. There may also be improvements in long-term survival associated with new antifibrotic drugs, as well as the implementation of new dietary and exercise strategies aimed at preserving cell mass and reversing the nutritional and metabolic risk situations associated with the disease.

Several factors influence the nutritional status of these patients. These include increased respiratory muscle strain, the release of inflammatory mediators, and the co-occurrence of hypoxemia and inactivity [6]. All of these impact the clinical outcomes of IPF, although there is little literature on the nutritional status and clinical variables relevant to the prognosis of their disease. Patients with IPF suffer from difficulty in exercising in addition to baseline physical inactivity and a deterioration in their quality of life; however, it has been described that almost 50% of patients with severe IPF and normal body weight had a nutritional deficiency, which means that nutritional assessment cannot be limited to weight or BMI and requires specific assessments to help us detect these nutritional problems [7].

In addition, disease progression and prognosis vary widely, depending on the presence of acute exacerbations, comorbidities, disease severity, and the availability and side effects of anti-fibrotic agents. Disease severity staging systems are critical and helpful in determining prognoses and guiding management choices. Several clinical predictive models have been developed for patients with IPF, and the most frequently used is the Gender, Age and Physiology (GAP) model [8].

Therefore, a new nutritional approach is needed that focuses on assessing nutritional status by looking at changes in composition and function, using parameters like phase angle (PhA) and other electrical measurements from bioelectrical impedance analysis (BIA), hand grip strength (HGS), functional testing, nutritional ultrasound (NU^®^), or laboratory values, such as C-reactive protein (CRP)/prealbumin [9]. The new approach to the morphofunctional assessment of DRM is the integration of classical parameters of nutritional assessment with emerging nutritional techniques that determine function and body composition.

One of these tools is the phase angle (PhA), which is measured via bioelectrical impedance analysis (BIA). It could be considered a good marker of a patient’s cell mass and cell damage. Several studies have shown that the PhA level is associated with increased nutritional risk in different pathologies [10,11,12,13]. Nevertheless, the use of PhA as a prognostic risk factor for mortality in patients with IPF has been the subject of only a few studies.

Nutritional Ultrasound^®^ (NU^®^) is a new approach that uses ultrasound measurements to identify and assess the thickness of the lean and fat layers. This allows the body composition (lean mass and fat mass) of the organism to be extrapolated. The body compartments of adipose, muscle, connective, vascular, and bone tissue are assessed using ultrasound. It is an emerging advanced clinical examination tool to assess the different body compartments, and there are various validation studies of the measurement technique [14]. Although different muscular structures can be assessed, most studies focus on the rectus femoris (RF), which is one of the most referenced measures for its correlation with strength and performance or functional tests [15,16].

Morphofunctional assessment has included the evaluation of functional tests, such as TUG and HGS, to determine the functional status and the effect of the prognosis of patients. The Timed Up and Go (TUG) test is recommended as a routine screening test for balance problems and falls in older adults. Some studies have shown that the TUG can be used to predict a history of falls and exercise capacity in COPD [17]. An impaired TUG was an independent predictor of increased 5-year mortality in older adults undergoing surgery for solid tumors [18].

The six-minute walk test (6MWT) is recommended in international guidelines because of its simplicity, and desaturation during 6MWT is a strong predictor of mortality in IPF patients [8].

Therefore, the aim of this study is to determine new predictors of mortality using a morphofunctional assessment of this group of patients. 

## 2. Materials and Methods

### 2.1. Setting Study

A prospective, observational, bicenter study of routine clinical practice was designed in the nutrition unit of the Endocrinology and Nutrition Unit of the Virgen de la Victoria University Hospital. Patients with idiopathic pulmonary fibrosis were diagnosed at different stages, taking advantage of the biannual assessment usually performed in the pneumology service of our hospital. Patients were received from our hospital and from the Regional University Hospital of Málaga.

All subjects gave their informed consent for inclusion before they participated in the study. The study was conducted in accordance with the Declaration of Helsinki and was approved by the Ethics Committee of Málaga on 5 April 2022. All patients enrolled in our study met the inclusion criteria (idiopathic pulmonary fibrosis and consent to participate in the study via accepted informed consent), and none of the exclusion criteria (refusal to participate or the inability to perform BIA measurement for reasons related to ethnicity, extensive skin lesions, the extravasation of fluids through the route and local hematomas, amputation, etc., or a life expectancy of fewer than 3 months). A flow chart diagram shows the patient selection process for our study (Appendix A).

### 2.2. Anthropometric and Body Composition Parameters

Body composition analysis was performed using a 50 kHz phase-sensitive impedance analyzer (BIA 101 Whole Body Bioimpedance Vector Analyzer (AKERN, Florence, Italy)) that delivered 800 μA [19,20] via tetrapolar electrodes placed on the right hand and foot. PhA was expressed in degrees as arctan (Xc/R) × (180°/π). An individual standardized PhA value (SPhA) was determined from the sex- and age-matched reference population value by subtracting the reference PhA value from the observed patient PhA value and dividing the result by the respective age- and sex-reference standard deviation (SD) [21].

Daily assessment of the technical accuracy of the BIA instrument used a precision circuit supplied by the BIA device manufacturer (Akern). All measured R and capacitance values were consistently ±1 Ω of the 385 Ohm reference value. We also determined the in vivo reproducibility of the BIA measurements and found coefficients of variation (CV) of 1–2% for R (resistance) and Xc (reactance). 

At each patient’s first visit, their body weight and standing height were measured. All BIA measurements were taken with the patient supine on a hospital bed. As fluid shifts occur when moving from a standing to a lying position and directly affect R and Z values, the patient remained in the supine position for five minutes before the BIA measurements were taken in order to stabilize the BIA values (±2 Ω for R and ±1 Ω for Xc). BIVA emphasizes the position of the impedance vector derived from R and Xc values normalized to height (H, m) on the R/Xc graph relative to tolerance ellipses generated from a sex-matched healthy reference population (e.g., 50, 75, and 97% versus 1, 2, and the SD) [20]. The BI measurements of the patients were standardized for sex and age using data from healthy Italian adults [19,22].

Hand grip strength (HGS) was also tested using a JAMAR hand dynamometer (Asimow Engineering Co., Los Angeles, CA, USA). Grip strength was measured in a seated position with the dominant hand elbow flexed at 90 degrees. Patients were instructed to perform three maximal isometric contractions with brief pauses between measurements, and the median value was recorded. 

Functional tests (Get Up and Go) were performed with the patient sitting in a chair and timing in seconds how long it took them to get up, walk 3 m, turn around, and walk another 3 m to sit down. The same was done with the 6-min walk [23], timing how many meters the patient traveled in 6 min to assess their ability.

### 2.3. Abdominal and Muscle Nutritional Ultrasound

Muscle ultrasonography of the quadriceps rectus femoris (QRF) of the lower extremity with a 10 to 12 MHz probe and a multifrequency linear matrix (Mindray Z50, Madrid, Spain) was performed with all subjects (each patient in the supine position). The evaluation was performed without compression at the level of the lower third from the superior pole of the patella and the anterior superior iliac spine, measuring the anteroposterior muscle thickness, circumference, and cross-sectional area [14]. The ultrasonography was performed by the same person who was trained in this technique previously.

The probe was oriented perpendicular to the longitudinal and transverse axes in the QRF, such as the rectus femoris cross-sectional area (RF-CSA), rectus femoris circunferencia (RF-CIR), RF-axis (-X-axis, and -Y-axis), and contracting muscle (RF-CON) and L-SAT (subcutaneous fat of the leg). The ultrasonography was performed by the same person who was familiar with the technique. Three measurements were performed for each parameter, and the mean was calculated.

In the abdomen, T-SAT (total subcutaneous abdominal fat), S-SAT (superficial subcutaneous abdominal fat), and VAT (preperitoneal or visceral fat) were measured in centimeters at the midpoint between the xiphoid appendix and the navel [24].

### 2.4. Assessment of Nutritional Status

Demographics, comorbidities, SGA, and clinical and anthropometric data were collected. A trained nutritionist classified patients into one of three classifications according to SGA: (A) well nourished, (B) moderately undernourished, or (C) severely undernourished. 

### 2.5. Assessment of Respiratory Status

Currently, the most commonly used prognostic index for IPF is the GAP (2). The GAP composite index was developed in 2012 and is based on the assessment of gender (G), age (A), and physiology (P), the latter including two conventional lung function parameters: forced vital capacity (FVC) and diffusing lung capacity for CO (DLCO) [25]. It gives a value for each parameter (sex: female = 0 vs. male = 1; age: ≤60 years = 0, 61–65 years = 1, >65 years = 2; % predicted FVC: >75% = 0, 50–75% = 1, <50% = 2; % predicted DLCO: >55% = 0, 36–55% = 1, ≤35% = 2, and unable to perform = 3), and the final sum score allows the estimation of the mean 1-, 2-, and 3-year mortality risk by disease stage (i.e., a score of 0–3 = stage I; a score of 4–5 = stage II; and a score 6–8 = stage III). The GAP has been validated with untreated IPF patients and, in recent publications, with treated patients [26].

### 2.6. Statistical Analysis

Data analysis was mainly performed using JAMOVI (version 2.2.2 MacOS). Descriptive statistics were used to characterize the patient cohort. The normality of the distribution of quantitative variables was checked using the Shapiro–Wilk test. Descriptive statistics were used to analyze categorical variables (absolute and relative frequencies) and quantitative variables (means and SDs or medians and interquartile ranges). Clinical data and BIA values between survival and non-survival were compared using Student’s t-test, the Mann–Whitney U test, or the Chi-squared test.

The assessment of the diagnostic performance of PhA, RF-CSA, and HGS in detecting the risk of death was based on receiver operating characteristic (ROC) curves and the area under the curve (AUC). We estimated the accuracy of these measurements using AUC by plotting sensitivity versus 1-specificity. ROC curves were used to determine the optimal cut-off values by finding the point of convergence for the greatest sensitivity and specificity. The area under the curve (AUC) indicates the discriminatory power of the test. Statistical significance was set to *p* < 0.05.

The Kaplan–Meier product-limit estimate at 12 months was used to calculate the accumulated probability of death, estimate survival, and assess the difference between the BIVA, NU, and functional test cut-points. Kaplan–Meier survival curves were compared using the log-rank (Mantel–Cox) test. The time of onset was the day of admission, the event was defined as death, and all cases were censored at the last observation. Differences were considered statistically significant at *p* < 0.05.

Cox regression with proportional hazards was used to assess the association of BIA and NU with mortality in IPF (idiopathic pulmonary fibrosis). Hazard ratios (HRs) and their 95% confidence intervals (CIs) were calculated. We used a multivariate model with RF-CSA and PhA, sex, age, and BMI.

We compared the values of PhA and RF-CSA with established prognostic indicators.

## 3. Results

### 3.1. General Characterization of the Population Study

A total of 86 patients with IPF were included. The mean age was 71.0 ± 7.26 years. A total of 71 patients were men (82.5%), and 15 were women (17.5%). We divided our population by mortality. Demographic data, anthropometric measures, functional tests, and patient outcomes are presented in Table 1. No significant difference between age and BMI or additional nutritional blood test parameters was found, but significant differences were found between the DLCO (higher in survival, 50.02 vs. 25.25, *p* < 0.001). However, we observed in BIVA parameters a significant difference in PhA (lower in non-survival, 4.85° vs. 4.27° *p* < 0.05) and FM (higher in non-survival, 23.2 vs. 29.0 kg, *p* <0.05). In terms of functional measurement, it was observed that non-survivors had a lower 6WMT (416 vs. 301 m, *p* < 0.001) and TUG (9.14 vs. 8.21 s, *p* < 0.05). And, finally, significant differences were observed according to the GAP stage in survival and non-survival. The differences between genders can be seen in Appendix A.

### 3.2. Correlation Analysis between BIA Muscle Measures, Muscular Ultrasound, HGS with 12-Month Mortality

Correlations existed between the classic body composition parameters of BIVA (BMI, FFMI, and FMI), BIVA (PhA and BCM), nutritional ultrasound (RF-CSA and RF-AXIS Y), and functional tests (HGS and TUG), which reflect the relationship between muscle and adipose tissue (Table 2). For this purpose, we performed a heat map correlation analysis (Figure 1). We observed that RF-CSA from nutritional ultrasound was strongly correlated with muscle measurements measured via BIVA. Of note were the correlations of RF-CSA with BCM (r = 0.70, *p* < 0.001), HGS (r = 0.54, *p* < 0.001), FFMI (r = 0.64, *p* < 0.001), and ASMI (r = 0.66, *p* < 0.001). The RF-Y-axis was also correlated with BCM (r = 0.64, *p* < 0.001) and FFMI (r = 0.67, *p* < 0.001). One of the functional parameters, HGS, showed a positive correlation with BIVA (BCM: r = 0.60, *p* < 0.001; ASMM: r = 0.61, *p* < 0.001; and FFM: r = 0.61, *p* < 0.001) and ultrasound parameters (RF-CSA: r = 0.54, *p* < 0.001; and RF-Y-axis: r = 0.46, *p* < 0.001) but little correlation with the classic parameter BMI. Similarly, the functional assessment via TUG did not show a significant relationship with any of the ultrasound data (RF-Y-axis: r = −0.03, *p* = 0.800) or BIVA muscle mass (SMI: r = −0.16, *p* = 0.161). We obtained a Cronbach’s scale of 0.778 between all techniques.

On the other hand, a correlation was made between the values of fat mass measured via BIVA and the adipose tissue measurement via nutritional ultrasound (Figure 2). It can be observed that the BMI value (the classically used parameter) correlates strongly with FMI-BIVA (r = 0.84, *p* < 0.001) and L-SAT (r = 0.40, <0.001) and weakly but significantly with T-SAT (r = 0.17, <0.001), S-SAT (r = 0.21, <0.001), and VAT (r = 0.19, <0.001).

### 3.3. Cut-Off Point for 12-Month Mortality in IPF Patients

To investigate the cut-off point parameters for predicting 12-month mortality in our population, we performed ROC analysis. We determined the cut-off point for 12-month mortality in IPF patients for BIVA, nutritional ultrasound, and functional tests (Table 3). We found a cut-off point of PhA = 4.5° with an AUC value of 0.722 (sensitivity of 72.7% and specificity of 66.6%). We also obtained a cut-off point of BCM = 28.8 kg with an AUC value of 0.609 (sensitivity of 32.4% and specificity of 100.0%), SPhA = −0.44 with an AUC value of 0.618 (sensitivity of 35.0% and specificity of 88.9%), and NaK with AUC = 0.562, cut-off = 1.17 (sensitivity of 66.67% and specificity of 53.25%). On the other hand, muscular nutritional ultrasound is also a good predictor of 12-month mortality (RF-CSA = 3.00 cm^2^ with an AUC value of 0.857, sensitivity of 64.41%, and specificity of 100.0% and RF-Y-axis = 1.10 with AUC = 0.615 (sensitivity of 47.37% and a specificity of 88.89%)) (Figure 3a). However, we could observe that the AUC values of the adipose tissue measurement of the nutritional ultrasound were not very decisive. We obtained a cut-off of T-SAT = 0.83 with an AUC = 0.474 (sensitivity of 94.94% and specificity of 22.22%) and S-SAT = 0.30 with an AUC = 0.437 (sensitivity of 97.37% and specificity of 11.11%); more importance can be given to the value of VAT with AUC = 0.658 (sensitivity of 62.50% and specificity of 72.97%). Finally, analyzing the functional parameters, we observed that the 6MMW with a cut-off of 420 m is a good determinant of 12-month mortality with AUC = 0.830 (sensitivity of 63.27% and specificity of 100.0%), as well as TUG with a cut-off of 7.2 s and AUC = 0.771 (sensitivity of 100.0% and specificity of 56.67%). However, the HGS shows a cut-off of 44 kg and AUC = 0.468 (sensitivity of 21.33% and specificity of 100.0%). Among the blood test parameters, we wanted to give preference to the CRP protein value, which is associated with inflammation at the time of mortality. We obtained a cut-off value of CRP protein = 7 g/dl with AUC = 0.731 (sensitivity of 100% and specificity of 55.56%) (Figure 3b).

### 3.4. 12-Month Mortality Risk for IPF Patients

We performed a multivariate analysis for the 12-month mortality of RF-CSA and PhA adjusted for age, gender, and BMI. We found that PhA–mortality has HR = 6.35 (1.29–31.15, *p* = 0.023) (Table 4); RF-CSA–mortality (HR = 8.11 (1.39–47.16, *p* = 0.020)) (Table 5) was independently associated with death (Figure 4A,B).

### 3.5. Kaplan–Meier Survival Curve of 12-Month Mortality in IPF Patients with Morphofunctional Assessment Techniques

The median follow-up time was 12 months, during which nine patients died. IPF patients with low RF-CSA (<3.00 cm^2^) died earlier (survival probability: 41.1% [19–90%, 95% CI]) than patients with normal RF-CSA (survival: 100% [100–100%]: OR = 40.71 [2.27 to 728.21, *p* = 0.011) (Figure 5A). 

The mortality rate of patients with low PhA (<4.5°) was higher (survival: 53.2% [31–92.5%, 95% CI]) than that of patients with normal PhA (95.0% [88–92.5%, 95% CI], OR = 5.48 [1.23–38.49, *p* = 0.042]) (Figure 5B). 

Patients with low BCM (<28.8 kg) achieved lower survival (survival probability: 69.3% [53−91.3%, 95% CI]) than patients with normal BCM (100% [100−100%, 95% CI], OR = 39.11 [2.1894 to 698.9], *p* = 0.0127) (Figure 5C).

Patients with low TUG (<7.2 s) achieved lower survival (survival probability: 97% [91−100.0%, 95% CI]) than patients with normal TUG (65% [43–96.9%, 95% CI], OR = 8.80 [1.41−170.71, *p* = 0.049] (Figure 5D).

Patients with low 6MWT (<420 m) achieved lower survival (survival probability: 60% [35−100%, 95% CI]) than patients with normal 6MWT (100% [100−100%, 95% CI], OR: 18.72 [0.9789 to 358.3, *p* = 0.051]) (Figure 5E).

Patients with GAP stage I achieved higher survival (survival probability: 100.0% [100–100%, 95% CI]) than patients with GAP stage II (63% [63−100%, 95% CI]) and stage III (23% [7–76%, 95% CI]; stage III vs. stage III, OR = 5.00 [0.79–31.63, *p* = 0.087]; stage I vs. stage II, OR = 0.71 [0.203–2.50, *p* = 0.599]; Stage II vs. stage III, OR = 7.00 [1.20–40.82, *p* = 0.0306]) (Figure 5F).

However, a low HGS value was not a determinant of 12-month mortality. Lower HGS was associated with lower survival (survival probability: 74% [58−92.7%, 95% CI]) than a normal HGS (100% [100−100%, 95% CI], OR 1.56 (0.28–12.00, *p* = 0.629)) (Figure 5G). The same applied to a lower BMI value, which was also associated with lower survival (survival probability: 33% [7–100%, 95% CI]), compared to patients with a normal BMI (84 [74%−96.8%, 95% CI], OR 0.58 (0.14–2.90, *p* = 0.460)).

## 4. Discussion

To the best of our knowledge, this is the first study that has related morphofunctional assessment techniques (PhA via BIVA, RF-CSA via nutritional ultrasound, and the functional tests 6MWT and TUG) with the risk of mortality in patients with idiopathic pulmonary fibrosis, establishing cut-off points from which we can predict a worse clinical evolution for our patients. 

Among these parameters, we obtained cut-off values of PhA < 4.6°, RF-CSA < 3.00 cm^2^, 6MWT < 420 m, and TUG > 7.2 s, which are among the best morphological parameters for predicting 12-month mortality.

This study included IPF patients with common characteristics of this disease, as they were mostly men with a normal BMI [27]. As described in the literature, a worse prognosis was found for patients with a higher GAP [28] and an altered functional test (6MWT) [29], with both parameters increasing mortality, which was related to the published results.

In our series, there were no differences between survival and non-survival patients’ general data, such as their age, sex, weight, weight loss, and BMI. Nakatsuka et al. found that >5% weight loss was associated with poor survival and that weight loss > 6.1^a^ was an independent predictor of mortality [3]. Other studies have examined the relationship between malnutrition and mortality for IPF patients by measuring BMI or weight loss, but BMI provided contradictory results [30]. Although BMI < 21 kg/m^2^ is used to define malnutrition in patients with chronic diseases, such as COPD [31], versus BMI < 25 for IPF patients [32], it is not sensitive enough to identify patients with a low muscle mass (FFM) in these studies.

However, there are clear differences in morphofunctional values with a decrease in PhA in the survival vs. non-survival groups (4.85° vs. 4.27° *p* = 0.033) according to BIVA. Gómez-Martínez et al. showed that low PhA (5.11° vs. 4.62°, *p* = 0.008) was independently associated with a worse prognosis for COPD patients [33]. Machado et al. showed that a quarter of IPF patients with a normal to obese BMI had abnormally low PhA, and patients with IPF classified as low-PhA had worse lung function, exercise capacity, and HRQL [34].

The other classical factors, GAP (*p* < 0.001) and DLCO (50.02 vs. 25.25, *p* < 0.001), showed significant differences in our series.

Our study adds to the current literature regarding the clinical applications of BIVA with bioelectrical values such as PhA in the prognosis of IPF patients. We found that the stratification of IPF patients according to their mortality risk is associated with a low PhA value. Previous studies have shown that high or low FFMI classification identified patients with significantly lower weight due to tissue depletion, including not only lower FFMI but also lower body cell mass and body water [34].

In this work, low PhA determined worse lung function in patients with IPF, as well as lower exercise capacity, despite no differences in BMI or other body composition variables (except for a higher amount of extracellular water). Increased extracellular water leads to a decrease in PhA, which is associated with a worse prognosis [35]. These findings support the use of PhA as a proxy indicator of cellular health (an increased cell number with improved membrane integrity and function) [36].

Regarding functional techniques, there were no differences in HGS, but there were differences in TUG (8.21 vs. 9.14 s, *p* < 0.019) and 6MWT (416 m vs. 301 m, *p* < 0.001). Mezquita et al. showed that a TUG time of 11.2 s had good sensitivity (0.75) and specificity (0.83) for identifying patients with a baseline 6-min walk distance <350 m, that the TUG is valid and responsive in COPD, and that an abnormal result is indicative of poor health outcomes. These differences in TUG, 6MWT, and GAP have been described in other papers [17,28,29].

There is an important correlation in the results of assessments of body composition using morphofunctional techniques for different pathologies [12,37]; thus, a correlation has been demonstrated between RF-CSA and PhA [13,15]. Also found in other publications were lines between more established techniques, such as BIVA (the correlation of RF-CSA with FFMI (r = 0.780; *p* < 0.001)) and new ultrasonography techniques (RF-CSA with HGS; r = 0.790; *p* < 0.001) [16]. These parameters relate what would be cellular and muscle mass with the importance of nutrition and the parameters of sarcopenia and muscle function. However, it is also important to highlight how morphofunctional techniques show good correlation when assessing adiposity classically represented through BMI, with good correlation of the content measured via BIVA (FMI) and abdominal ultrasound (L-SAT, T-SAT, S-SAT, and VAT). This aspect of excess fat assessment is important, as IPF combines overweight and fat gain with a disease-related decrease in muscle mass [38].

These aspects are very interesting, as they are techniques of different natures with ultrasound, bioelectrical, and functional techniques with an adequate clinical correlation that allows them to be evaluated together when establishing diagnostic and prognostic tools. Their future role in the diagnosis of malnutrition and other problems affecting body composition and function, such as obesity and metabolic diseases, remains to be definitively established on the basis of future publications [37].

In addition, the establishment of cut-off-point predictors of mortality to assess these patients may have important clinical value. The nutritional ultrasound technique yields an RF-CSA value with adequate sensitivity and specificity to predict 12-month mortality. Other techniques, such as CT [39] or MRI [40], have already established their prognostic value for IPF patients and other pulmonary diseases [41]. De Paula et al. [40] have obtained preliminary findings suggesting that other factors, such as hypoxia (but not inflammation), may play a role in the peripheral skeletal muscle dysfunction observed in IPF patients, in addition to disuse atrophy. 

These imaging techniques are expensive and can only be used opportunistically to determine muscle mass. It is, therefore, necessary to establish routine clinical practice techniques, such as ultrasound, that can be performed at the point of patient care in a flexible, individualized, and repeated manner over time. Nutritional ultrasound techniques to assess muscle mass and adipose tissue have been used to diagnose nutritional status and metabolic disorders such as obesity. Point-of-care ultrasound (POCUS) [42] is an advanced diagnostic ultrasound test performed and interpreted by the treating physician as a bedside test. There are several advantages to incorporating POCUS into daily clinical practice. POCUS is a cost-effective approach that directly and indirectly saves healthcare costs on an international scale [43].

Many factors contribute to reduced muscle mass in IPF patients, including aging, smoking, reduced caloric intake, increased respiratory muscle demand, reduced physical activity, systemic inflammation, and the side effects of steroid and anti-fibrotic treatments [6]. 

On the other hand, BIVA is a widely used instrument in clinical practice and has been of particular interest in recent years for bioelectrical measurements. The phase angle in health and disease plays an increasing role in lines of research [15].

Among the absolute variables detected via BIA, the phase angle (PhA) represents an interesting parameter for assessing the state of health of patients with respiratory diseases. In contrast, easier methods that provide indirect information, such as BIA, are susceptible to errors due to changes in the hydration of a fat-free mass. The use of raw BIVA measurements avoids inherent assumptions. PhA is a basic bioimpedance measurement that provides a qualitative index of fluid status and body cell mass or lean soft tissue mass.

In our study, the RF-CSA and PhA model showed a good predictive ability for 12-month mortality adjusted for age, sex, and BMI. Other factors, such as BCM, TUG, 6MWT, and GAP, showed a determinant value for predicting mortality.

In light of our findings, RF-CSA and PhA values could be employed in the routine clinical evaluation of IPF patients for better prognostic risk stratification of IPF patients.

Nevertheless, this study faced limitations. All the patients were from only two areas, so the results need to be confirmed for other IPF populations. Second, we did not have analytical data with which to assess systemic inflammation or mitochondrial damage, but we collected these samples for the second phase of the study; this could have slightly influenced our results. Finally, we did not record the causes of death. Large-sample, multi-center studies involving more female patients are required in the future to clarify the differences in our findings.

## 5. Conclusions

In summary, PhA via BIVA and RF-CSA via nutritional point-of-care ultrasound (nutritional POSE) can accurately predict the 12-month mortality measurements of IPF patients, which could guide the care process for more individualized treatment decisions, nutritional support, exercise, etc. There is a need to identify new biomarkers to support classical prognostic factors, such as GAP and T6MW. Since PhA and RF-CSA are identifiers of nutritional and muscular impairment, this is relevant to address the prognosis of IPF patients with the aim of improving their quality of life and survival. Future studies will determine whether a dedicated intervention could improve outcomes for IPF patients.

## Figures and Tables

**Figure 1 nutrients-15-04473-f001:**
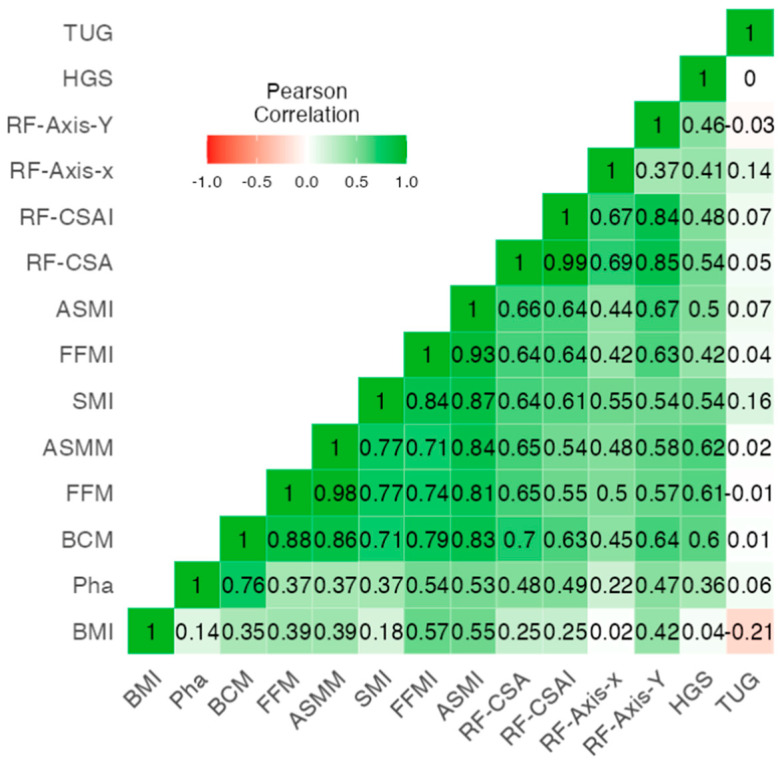
Pearson’s correlation heatmap between the BIVA muscular parameters, muscle nutritional ultrasound, and HGS. Abbreviations: BMI, body mass index; PhA, phase angle; BCM, body mass cell; SMI, skeletal mass index; FFMI, fat-free mass index; ASMI, appendicular skeletal muscle index; HGS, hand grip strength; RF-CSA, rectus femoris cross-sectional area; TUG, Get Up and Go test.

**Figure 2 nutrients-15-04473-f002:**
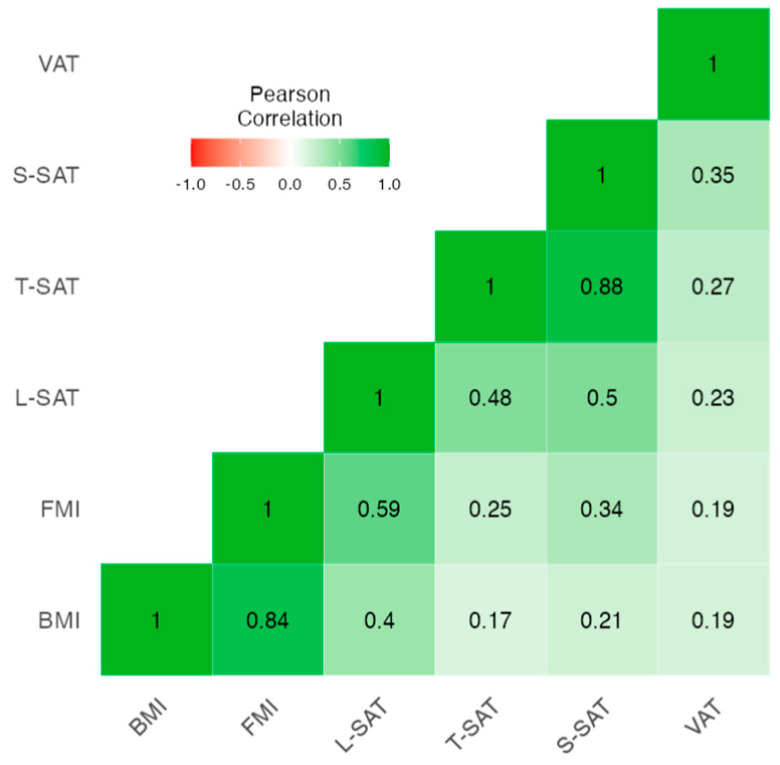
Pearson’s correlation heatmap between the BIVA fat measures and abdominal nutritional ultrasound. Abbreviations: BMI, body mass index; FMI, fat mass index; L-SAT, leg subcutaneous adipose tissue, SAT, abdominal adipose tissue, total (T) and superficial (S); VAT, visceral adipose tissue.

**Figure 3 nutrients-15-04473-f003:**
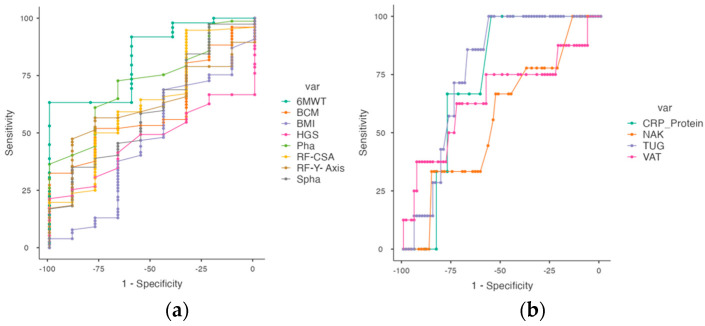
Comparative analysis of ROC curve of BIVA measurements (BMI, BCM, PhA, SPhA, and NaK), nutritional ultrasound (RF-CSA and RF-Y-axis), blood test (CRP protein), and functional tests (HGS, 6MWT, and TUG) with established cut-off for 12-month mortality in IPF patients. Abbreviations: BMI, body mass index; RF-CSA, rectus femoris cross-sectional area; RF-Y-axis, rectus femoris Y-axis; VAT, visceral adipose tissue; BIVA, bioelectrical impedance vectorial analysis; PhA, phase angle; SPhA, standardized phase angle; BCM, body mass cell; NaK, sodium potassium index; HGS, hand grip strength; TUG, Get Up and Go test; 6MWT, 6-min walk test. (**a**) morphofunctional parameters in IPF patients. (**b**) Inflammation parameters (NaK, CRP protein, and VAT) and Get Up and Go test in IPF patients.

**Figure 4 nutrients-15-04473-f004:**
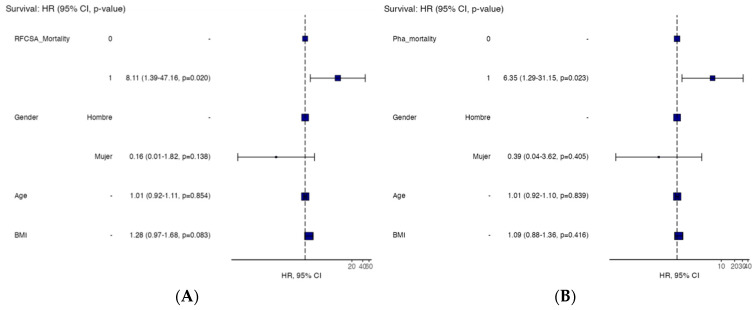
Hazard ratio multivariable analyses by BMI, age, gender, and PhA–mortality (**A**) and RF-CSA–mortality (**B**) in IPF patients. Abbreviations: RF-CSA, rectus femoris cross-sectional area; PhA, phase angle; BMI, body mass index.

**Figure 5 nutrients-15-04473-f005:**
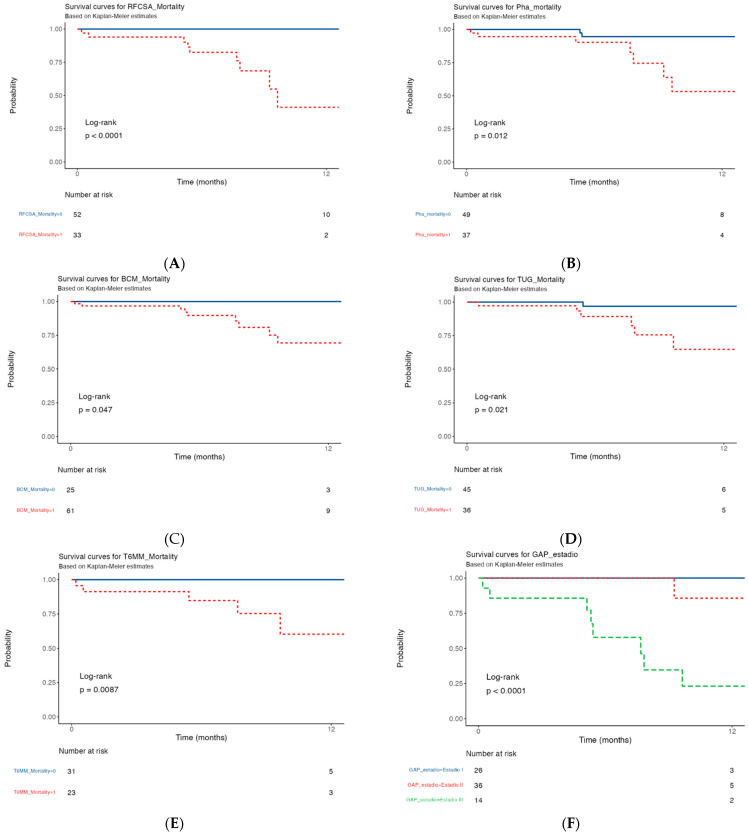
Kaplan–Meier survival curves of IPF patients with low or normal values in groups by classical parameters (GAP and T6MM) vs. new parameters (RF-CSA, PhA, BCM, and TUG–mortality). (**A**) RF-CSA–mortality; (**B**) PhA–mortality; (**C**) BCM–mortality; (**D**) TUG–mortality; (**E**) 6MWT–mortality; (**F**) GAP stage; (**G**) HGS–mortality; (**H**) BMI–mortality. Low-value patients (red line) died more frequently than normal-PhA–mortality-value patients (blue line). Abbreviations: RF-CSA, rectus femoris cross-sectional area; PhA, phase angle; BCM, body mass cell; HGS, hand grip strength; TUG, Get Up and Go test; 6MWT, 6-min walk test; HGS mean, mean hand grip strength; GAP, gender (G), age (A), and physiology (P) index; BMI, body mass index.

**Table 1 nutrients-15-04473-t001:** Baseline characteristics of the population of study divided 12-month mortality.

All		Survival	Non-Survival	*p*-Value
	*N* = 86	*N* = 77	*N* = 9	
**Demographic variables**				
Age (years)	71.0 (7.26)	71.0 (7.37)	71.0 (6.65)	0.949
Weight (kg)	78.30 (12.1)	77.7 (11.7)	83.9 (14.6)	0.144
Weight loss (%)	6.41 (6.78)	6.26 (6.66)	7.71 (8.12)	0.696
BMI (kg/m^2^)	27.40 (3.44)	27.3 (3.46)	28.0 (3.41)	0.612
**BIA**				
PhA (°)	4.78 (0.77)	**4.85 (0.76)**	**4.27 (0.65)**	**0.033**
SPhA	−1.03 (1.01)	−0.98 (0.99)	−1.41 (1.06)	0.23
Hydration (%)	74.60(2.31)	74.6 (2.36)	75.1 (1.87)	0.165
NaK	1.18 (0.18)	1.17 (0.18)	1.21 (0.16)	0.548
BCM (kg)	25.5 (5.17)	25.7 (5.28)	23.8 (3.89)	0.287
FFM (kg)	54.5 (7.48)	54.47 (7.61)	54.98 (6.49)	0.868
ASMM (kg)	20.2 (3.29)	20.17 (3.36)	20.38 (2.77)	0.383
SMI (cm^2^/m^2^)	8.81 (1.20)	8.85 (1.22)	8.41(1.06)	0.299
FFMI (%)	19.0 (1.74)	19.1 (1.74)	18.4 (1.70)	0.219
FM (kg)	23.8 (7.91)	**23.2 (7.37)**	**29.0 (10.8)**	**0.038**
**Echography exploration**				
RF-CSA (cm^2^)	3.38 (0.98)	3.43 (0.99)	2.96 (0.84)	0.196
RF-CIR (cm)	8.15 (1.11)	8.19 (1.10)	7.81 (1.23)	0.333
RF-X-axis (cm)	3.43 (0.50)	3.45 (0.50)	3.32 (0.49)	0.466
RF-Y-axis (cm)	1.11 (0.27)	1.12 (0.28)	1.02 (0.16)	0.291
L-SAT (cm)	0.78 (0.52)	0.78 (0.53)	0.78 (0.44)	0.563
T-SAT (cm^2^)	1.67 (0.71)	1.66 (0.71)	1.73 (0.77)	0.779
S-SAT (cm^2^)	0.72 (0.30)	0.71 (0.29)	0.79 (0.35)	0.544
VAT (cm^2^)	0.65 (0.30)	0.62 (0.25)	0.89 (0.56)	0.146
**Functional measurement**				
HGS max (kg)	34.5 (10.4)	34.4 (10.8)	36.0 (5.61)	0.655
HGS mean (kg)	33.0 (10.1)	32.9 (10.5)	34.0 (5.61)	0.761
TUG (s)	8.29 (5.24)	**8.21 (5.45)**	**9.14 (1.93)**	**0.019**
6MW (m)	405.0 (76.2)	**416 (61.6)**	**301 (128)**	**<0.001**
**Biochemical variables**				
Glucose (mg/dL)	110 (38.3)	111 (27.5)	105 (80.1)	**0.038**
Urea (mg/dL)	43.5 (14.8)	42.6 (14.0)	47.0 (18.7)	0.567
Creatinine (mg/dL)	1.05 (0.25)	1.08 (0.26)	0.90 (0.10)	0.165
Total cholesterol (mg/dL)	198 (58.8)	198 (56.3)	199 (85.0)	0.985
Triglycerides (mg/dL)	142 (90.5)	131 (89.5)	195 (91.9)	0.274
FCV (%)	67.9 (15.9)	68.5 (16.5)	62.1 (9.49)	0.265
FEV1 (%)	77.9 (19.4)	78.8 (20.5)	10.0 (5.40)	0.233
DLCO (%)	47.3 (18.2)	**50.02 (17.1)**	**25.25 (10.7)**	**<0.001**
**Clinicopathological variables**				
Diagnostic (month)	15.5 (19.2)	16.2 (19.8)	10.1 (11.8)	0.436
GAP Stage:				**<0.001**
I	26 (34.2%)	26.0 (34.2%)	0.0 (0.00%)	
II	36 (47.4%)	35.0 (46.1%)	1.0 (1.30%)	
III	14 (18.4%)	6.0 (7.9%)	8.0 (10.5%)	
SGA				0.396
A	15.0 (17.4%)	15.0 (17.4%)	0.0 (0.0%)	
B	52.0 (60.5%)	45.0 (52.3%)	7.0 (8.1%)	
C	19.0 (22.1%)	17.0 (19.8%)	2.0 (2.3%)	

Data are expressed as means ± standard deviations or percentages. Groups were divided by the 12-month mortality variables. Abbreviations: BMI, body mass index; BIVA, bioelectrical impedance vectorial analysis; PhA, phase angle; SPhA, standardized phase angle; NaK, sodium potassium index; BCM, body mass cell; FFM, fat-free mass; ASMM, appendicular skeletal mass muscle; SMI, skeletal mass index; FFMI, fat-free mass index; FM, fat mass; HGS mean, mean hand grip strength; HGS max, maximum hand grip strength; TUG, Get Up and Go test; 6MWT, 6-min walk test; RF- CIR, circumference of the quadriceps rectus femoris; RF-CSA, rectus femoris cross-sectional area; RF-X-axis, rectus femoris X-axis; RF-Y-axis, rectus femoris Y-axis; L-SAT, leg subcutaneous adipose tissue; SAT, abdominal subcutaneous adipose tissue, total (T) and superficial (S); VAT, visceral adipose tissue; FCV, forced vital capacity; FEV1, forced expiratory volume in the first second; DLCO, pulmonary carbon monoxide diffusing capacity; GAP, gender (G), age (A), and physiology (P) index; SGA, subjective global assessment.

**Table 2 nutrients-15-04473-t002:** Correlation between BIA muscle measures, nutritional ultrasound, and HGS.

	RF-CSA	RF-CSAI	RF-X-Axis	RF-Y-Axis	HGS	TUG
BMI (kg/m^2^)						
	r= 0.25	r = −0.25	r = −0.02	r = 0.42	r = 0.04	r = −0.21
	*p* < 0.05	*p* = 0.056	*p* = 0.885	*p* < 0.001	*p* = 0.844	*p* < 0.211
PhA (°)	r = 0.48	r = 0.49	r = 0.22	r = 0.47	r = 0.348	r = −0.06
	*p* < 0.001	*p* < 0.001	*p* = 0.096	*p* < 0.001	*p* < 0.001	*p* = 0.620
BCM(kg)	r= 0.70	r = 0.63	r= 0.45	r = 0.64	r = 0.60	r = 0.01
	*p* < 0.001	*p* = <0.001	*p* < 0.001	*p* < 0.001	*p* < 0.001	*p* = 912
FFM(kg)	r = 0.65	r = 0.55	r = 0.50	r = 0.57	r = 0.61	r = −0.01
	*p* < 0.001	*p* < 0.001	*p* < 0.001	*p* < 0.001	*p* < 0.001	*p* = 920
ASMM (kg)	r = 0.65	r = 0.54	r = 0.48	r = 0.58	r = 0.62	r = 0.02
	*p* < 0.001	*p* < 0.001	*p* < 0.001	*p* < 0.001	*p* < 0.001	*p* = 850
SMI (kg/m)	r = 0.64	r = 0.61	r = 0.55	r = 0.54	r = 0.54	r = 0.16
	*p* < 0.001	*p* < 0.001	*p* < 0.001	*p* < 0.001	*p* < 0.001	*p* = 0.161
FFMI(Kg/m)	r = 0.64	r = 0.64	r = 0.42	r = 0.63	r = 0.42	r = 0.04
	*p* < 0.001	*p* < 0.001	*p* < 0.001	*p* < 0.001	*p* < 0.001	*p* < 0.739
ASMI (Kg/m)	r = 0.66	r = 0.64	r = 0.44	r = 0.67	r = 0.50	r = 0.07
	*p* < 0.001	*p* < 0.001	*p* < 0.001	*p* < 0.001	*p* < 0.001	*p* < 0.800
Handgrip strength (kg)	r = 0.54	r = 0.54	r = 0.41	r = 0.46	--	r = −0.358
	*p* < 0.001	*p* < 0.001	*p* < 0.001	*p* < 0.001		*p* < 0.001
TUG	r = 0.05	r = 0.07	r = 0.14	r = −0.03	r = −0.358	--
	*p* < 0.651	*p* < 0.555	*p* < 0.633	*p* < 0.800	*p* < 0.001	

Abbreviations: BMI, body mass index; BIVA, bioelectrical impedance vectorial analysis; PhA, phase angle; BCM, body mass cell; FFM, fat-free mass; ASMM, appendicular skeletal mass muscle; SMI, skeletal mass index; FFMI, fat-free mass index; ASMI, appendicular skeletal muscle index; HGS, hand grip strength; TUG, Get Up and Go test; RF-CSA, rectus femoris cross-sectional area: RF-CSAI, rectus femoris cross-sectional area index; RF-X-axis, rectus femoris X-axis; RF-Y-axis, rectus femoris Y-axis; HGS, hand grip strength.

**Table 3 nutrients-15-04473-t003:** Predictive value of 12-month mortality with BIVA, nutritional ultrasound, and functional tests of IPF patients.

	AUC	Cut-Off ▴	Sensitivity	Specificity
**Rectus Femoris**				
RF-CSA	0.857	3.00	64.41%	100.0%
RF-CIR	0.577	8.79	35.53%	88.89%
RF-X-axis	0.567	3.88	22.37%	100.0%
RF-Y-axis	0.615	1.10	47.37%	88.89%
L-SAT	0.440	0.65	42.67%	66.67%
**Abdominal**				
T-SAT	0.474	0.83	94,94%	22.22%
S-SAT	0.437	0.30	97.37%	11.11%
VAT	0.658	0.75	62.50%	72.97%
* **BIA** *				
SPhA	0.618	−0.44	35.0%	88.9%
PhA	0.722	4.5	72.7%	66.6%
BCM	0.609	28.8	32.47%	100.0%
NaK	0.562	1.17	66.67%	53.25%
**Functional test**				
HGS	0.468	44.0	21.33%	100.0%
TUG	0.771	7.20	100.0%	56.76%
6MM	0.830	420.0	63.27%	100.0%
**Blood test**				
CRP protein	0.731	7	100%	55.56%

Receiver operating characteristic (ROC) for the nutritional ultrasound, BIA, and functional tests with the 12-month mortality of IPF patients. AUC, the area under the ROC curve. ▴ cut-off without adjusting for variables. Abbreviations: AUC, area under curve; RF-CSA, rectus femoris cross-sectional area: RF-CIR, circumference of quadriceps rectus femoris; RF-X-axis; rectus femoris X-axis; RF-Y-axis, rectus femoris Y-axis; L-SAT, leg subcutaneous adipose tissue; SAT, abdominal subcutaneous adipose tissue, total (T) and superficial (S); VAT, visceral adipose tissue; BIVA, bioelectrical impedance vectorial analysis; PhA, phase angle; SPhA, standardized phase angle; BCM, body mass cell; NaK, sodium potassium index; HGS, hand grip strength; TUG, Get Up and Go test; 6MWT, 6-min walk test.

**Table 4 nutrients-15-04473-t004:** Model multivariate analysis to evaluate the utility of PhA as a prognostic indicator of 12-month mortality in IPF patients.

Dependent: Survival (My Time, My Outcome)		All	HR (Univariable)	HR (Multivariable)
PhA–mortality	Survival	49 (57.0)	-	-
	Non-survival	37 (43.0)	5.92 (1.23–28.55, *p* = 0.027)	6.35 (1.29–31.15, *p* = 0.023)
Gender	Male	71 (82.6)	-	-
	Female	15 (17.4)	0.67 (0.08–5.38, *p* = 0.706)	0.39 (0.04–3.62, *p* = 0.405)
Age	Mean (SD)	71.0 (7.3)	1.02 (0.93–1.12, *p* = 0.669)	1.01 (0.88–1.36, *p* = 0.416)
BMI	Mean (SD)	27.4 (3.4)	1.06 (0.87–1.28, *p* = 0.579)	1.09 (0.88–1.36, *p* = 0.416)

Abbreviations: HR, hazard ratio; PhA, phase angle; BMI, body mass index.

**Table 5 nutrients-15-04473-t005:** Model multivariate analysis to evaluate the utility of RF-CSA as a prognostic indicator of 12-month mortality in IPF patients.

Dependent: Survival (My Time, My Outcome)		All	HR (Univariable)	HR (Multivariable)
RF-CSA–mortality	Survival	40 (47.1)	-	-
	Non-survival	45 (52.9)	3.92 (0.81–18.97, *p* = 0.089)	8.11 (1.39–47.16, *p* = 0.020)
Gender	Male	70 (82.4)	-	-
	Female	15 (17.6)	0.66 (0.08–5.34, *p* = 0.700)	0.16 (0.01–1.82, *p* = 0.138)
Age	Mean (SD)	70.9 (7.3)	1.02 (0.93–1.12, *p* = 0.655)	1.01 (0.92–1.11, *p* = 0.854)
BMI	Mean (SD)	27.3 (3.2)	1.07 (0.87–1.31, *p* = 0.521)	1.28 (0.97–1.68, *p* = 0.083)

Abbreviations: HR, hazard ratio; RF-CSA, rectus femoris cross-sectional area; BMI, body mass index.

## Data Availability

The data that support the findings of this study are available from the corresponding author upon reasonable request.

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
