# Peer review of "Rectus Femoris Cross-Sectional Area and Phase Angle asPredictors of 12-Month Mortality in Idiopathic Pulmonary Fibrosis Patients"

_nutrients, 2023, doi:10.3390/nu15204473_

Round 1
Reviewer 1 Report
A well-designed and written study. However, there are some points to improve:
- Please add the name of the review board, its location, approval number, and date of approval to your manuscript.
- How were the patients selected (e.g., consecutively, randomly, or selectively)?
- The process of randomization and allocation concealment has not been described.
- Were no patients lost to follow-up? If so, please state it.
- Have you performed power analysis prior to the study? How was the sample size determined?
- The results section is a bit exhausting; it is better shortened just to focus on the main results and avoid repetition of the table content.
- Abbreviations should be explained as subtitles below the tables.
Minor editing of the English language is necessary to ensure a polished and error-free manuscript.
Author Response
Dear Editor,
Please, find enclosed a reviewed version of our manuscript, “Rectus femoris cross-sectional area and phase angle as a predictor of 12 month-mortality in idiopathic pulmonary fibrosis patients.” We would like to thank the Editorial Team for giving us the opportunity to revise and improve our manuscript, and the Reviewers for their thoughtful and constructive comments. We have considered all the suggestions and have incorporated them into the revised manuscript. Changes to the original manuscript are highlighted in yellow. A point-by-point response to the Reviewers’ comments is provided below.
COMMENTS FROM NUTRIENTS REVIEWERS
Point 1: Please add the name of the review board, its location, approval number, and date of approval to your manuscript.
Response 1: The study was conducted in accordance with the Declaration of Helsinki and was approved by the Ethics Committee of Malaga on 5 April 2022 (1743-N-21). (I have already modified it on line 130 of the manuscript)
Point 2: How were the patients selected (e.g., consecutively, randomly, or selectively)?
Response 2: The patients were selected consecutively, all patients with IPF in Malaga (Hospital universitario virgen de la victoria and the Regional de malaga) were assessed and referred by the pneumologists to our practice.
Point 3: The process of randomization and allocation concealment has not been described.
Response 3: As mentioned in the previous question, all patients were assessed, regardless of their time of diagnosis. The idea was to explore the pathology with all cases in our province.
Point 4: Were no patients lost to follow-up? If so, please state it.
Response 4: In this article we present data from the baseline sample, we only follow up mortality at one year. In future articles we would like to show data from the follow-up at 6 months and 1 year. Of couse, we had some losses but more than 80% we managed to follow up due to the good cooperation of the patients.
Point 5: Have you performed power analysis prior to the study? How was the sample size determined?
Response 5: Yes, before starting the project, a sample analysis was carried out and it was determined that, given the low prevalence of the disease, approximately 70 patients followed in the clinic would be sufficient, as it is a rare disease. An alpha risk of 5% and a beta risk of 10% in a bilateral contrast were accepted, a minimum of 52 patients were required and a rate of missing or invalid patients was estimated at 5%.
Points 6: The results section is a bit exhausting; it is better shortened just to focus on the main results and avoid repetition of the table content.
Response 6: I think it's great, but if you tell me which of the results you are referring to, I'll reduce them. I assume you are referring to the kaplan meier curves, we could focus only on the significant ones, and the rest we can leave in supplementary if you agree?
Points 7: Abbreviations should be explained as subtitles below the tables.
Response 7: Perfect, I will modify them in the manuscript (red letter).

Reviewer 2 Report
The authors presented the study to investigate different morphofunctional techniques in the prognostic of mortality in patients with IPF. The method is straightforward and the figures are easy to follow. This paper addressed a significant clinical treatment. The paper is well written, and I only have couple comments:
1. In Table 1, is glucose considered statistically significant? If so, why not highlighted it?
2. In discussion,
a. Line 410, “it is not sensitive enough to identify patients with a low muscle mass (FFM) in these studies”. Please explain reason for the contradictory results from your study as well as the other literatures.
b. Line 429, the citation is missing.
c. T-SAT and S-SAT were shown with high sensitivity for the predictive value. Please explain.
Please correct minor errors and spellings
Author Response
Dear Editor,
Please, find enclosed a reviewed version of our manuscript, “Rectus femoris cross-sectional area and phase angle as a predictor of 12 month-mortality in idiopathic pulmonary fibrosis patients.” We would like to thank the Editorial Team for giving us the opportunity to revise and improve our manuscript, and the Reviewers for their thoughtful and constructive comments. We have considered all the suggestions and have incorporated them into the revised manuscript. Changes to the original manuscript are highlighted in red. A point-by-point response to the Reviewers’ comments is provided below.
Point 1: In Table 1, is glucose considered statistically significant? If so, why not highlighted it?
Response 1: You were right, I have modified it in table 1.
Point 2: 2. In discussion,
- Line 410, “it is not sensitive enough to identify patients with a low muscle mass (FFM) in these studies”. Please explain reason for the contradictory results from your study as well as the other literatures.
- Line 429, the citation is missing.
- T-SAT and S-SAT were shown with high sensitivity for the predictive value. Please explain.
Response 2:
- One of the classical prognostic assessment techniques (GAP) in this pathology takes into account age, sex and activity level. In the literature we saw how the inclusion of BMI could improve this prediction of mortality risk. In our study we focused on new prognostic assessment techniques such as BIVA and nutritional ultrasound which we believe that together with age, sex and BMI could give a more accurate and diagnostic view of these patients, by including the measurement of muscle mass. In the future it will be necessary to include these measurements to make medicine more accurate.
- I think it was a mistake and there was no citation there. It is modified in the text.
- In the line 455-460 we want to explain that the new techniques that we show in addition to accurately measure the cells (muscle mass) also give us data concerning the fat measured with BIVA and nutritional ultrasound (FM, L-SAT, S-SAT, T-SAT, VAT) and that correlates with BMI, i.e. that the BMI may have losses when detecting sarcopenia in these patients due to an overestimation of the weight from the fat.
